

# Assessing acute effects of two motor-cognitive training modalities on cognitive functions, postural control, and gait stability in older adults: a randomized crossover study

Ran Li[1,*], Ping Qu[2,*], Xue Hu[3], Xiaojing Li[1], Haiqing Zeng[1], Binghong Gao[3] and Zhiyuan Sun[1]

[1] School of Exercise and Health, Shandong Sport University, Jinan, China
[2] Department of Physical Education, Sun Yat-sen University, Guangzhou, China
[3] School of Athletic Performance, Shanghai University of Sport, Shanghai, China
* These authors contributed equally to this work.

## ABSTRACT

**Background:** The process of aging often accompanies a decline in cognitive function, postural control, and gait stability, consequently increasing the susceptibility to falls among older individuals. In response to these challenges, motor-cognitive training has emerged as a potential intervention to mitigate age-related declines.

**Objective:** This study aims to assess the acute effects of two distinct motor-cognitive training modalities, treadmill dual-task training (TMDT) and interactive motor-cognitive training (IMCT), on cognitive function, postural control, walking ability, and dual-task performance in the elderly population.

**Method:** In this randomized crossover study, 35 healthy elderly individuals (aged 60–75) participated in three acute training sessions involving TMDT, IMCT, and a control reading condition. Assessments of executive function, postural control, gait performance, and cognitive accuracy were conducted both before and after each session.

**Results:** Both TMDT and IMCT improved executive functions. Notably, IMCT resulted in a significant enhancement in correct response rates and a reduction in reaction times in the Stroop task ($p < 0.05$) compared to TMDT and the control condition. IMCT also led to an increase in dual-task gait speed ($p < 0.001$) and showed a trend towards improved cognitive accuracy ($p = 0.07$). Conversely, TMDT increased postural sway with eyes open ($p = 0.013$), indicating a potential detriment to postural control.

**Conclusion:** The findings suggest that IMCT holds greater immediate efficacy in enhancing cognitive function and gait stability among older adults compared to TMDT, with a lesser adverse impact on postural control. This underscores the potential of IMCT as a preferred approach for mitigating fall risk and enhancing both cognitive and physical functions in the elderly population.

Corresponding authors
Binghong Gao,
binghong.gao@hotmail.com
Zhiyuan Sun, 1213225245@qq.com

## INTRODUCTION

As individuals age, their skeletal muscle and sensory systems undergo degenerative changes that can lead to a decline in balance ability and gait stability. Consequently, this raises the risk of falls and instability among older adults. Numerous randomized controlled trials have demonstrated that targeted exercise regimens can effectively enhance various physical functions, including muscle strength (*Joshua, 2014*; *Clemson et al., 2012*), postural control (*Pata, Lord & Lamb, 2014*; *Telli & Cavlak, 2022*), and gait stability (*Freiberger et al., 2012*), thus mitigating the risk of falls. Moreover, aging is often accompanied by a decline in cognitive processes such as working memory and attention, as observed by *Van Diest et al. (2013)*. This decline in attention and working memory, combined with heightened attention demands for postural control, contributes to the challenge older adults face in maintaining their posture. Consequently, when older individuals are tasked with dual tasks, such as walking while talking, their movement stability diminishes, thereby increasing the risk of falls (*Jehu, Paquet & Lajoie, 2017*; *Van Diest et al., 2013*). Academics have shown considerable interest in investigating whether dual-task training, particularly motor-cognitive dual-task training, could result in more significant improvements in physical and cognitive functions compared to exercise or cognitive training carried out independently (*Tait et al., 2017*). Motor-cognitive training involves simultaneously engaging in both cognitive and motor training activities (*Lauenroth, Ioannidis & Teichmann, 2015*). Several studies have highlighted the significance of motor-cognitive training in enhancing gait stability (*Pitta et al., 2020*; *Rodacki et al., 2021*; *Schättin et al., 2016*), balance (*Teraz et al., 2022*), functional physical (*Rica et al., 2020*), and executive functions fitness (*Eggenberger et al., 2016*; *Schoene et al., 2015*) among the elderly.

In recent studies investigating the key factors contributing to the effectiveness of motor-cognitive training, researchers are increasingly acknowledging that different combinations of motor and cognitive tasks can lead to varied alterations in neuroplasticity. *Herold et al. (2018)* categorized simultaneous motor-cognitive training into two groups: (I) motor-cognitive training with additional cognitive tasks and (II) motor-cognitive training with incorporated cognitive tasks. The former resembles classic dual-task training, where the cognitive task is unrelated to the motor task and acts as a distractor (*e.g.*, reciting the alphabet backward while riding a bicycle) (*Herold et al., 2018*). The latter involves integrating the cognitive task into the motor task, where the cognitive component serves as a relevant prerequisite for completing the motor-cognitive task (*e.g.*, virtual reality bicycle exergames). The researchers argued that integrating cognitive tasks into motor tasks is more beneficial for stabilizing neuroplasticity effects, compared to using cognitive tasks as mere distractions (*Herold et al., 2018*). Numerous studies support this claim. For instance, research has shown that integrating cognitive tasks into physical activities improved numeracy skills, perceived enjoyment of learning (*Mavilidi et al., 2018*), and foreign

language vocabulary (*Toumpaniari et al., 2015*) in preschool children more effectively than engaging in physical activities unrelated to cognitive tasks.

However, there is a scarcity of research comparing the effects of various motor-cognitive training modalities on cognitive and physical functions related to fall risk among older adults. Additionally, there is a lack of randomized controlled experimental designs to evaluate the immediate impacts of motor-cognitive training on cognition and postural control in this population. The acquisition of acute data in this context is crucial for the following reasons.

Firstly, studies have demonstrated that short bursts of exercise can impact both static and dynamic balance performance among the elderly, which raises questions about the overall effects of such exercise. For example, researchers have observed that submaximal cycling has been linked to impaired postural control and gait stability (*Hamacher et al., 2016*; *Stemplewski et al., 2012*). These findings suggest a potential increased risk of falls following submaximal aerobic exercise. However, the immediate effects of motor-cognitive training involving lower limb movements on gait and postural control remain uncertain. Given that motor-cognitive training is often recommended to improve physical and cognitive functions and prevent falls among the elderly (*Schoene et al., 2014*), understanding these acute effects could raise awareness, address safety concerns, and facilitate a more informed approach to their physical training.

Secondly, research has shown that short-term aerobic exercise and high-intensity endurance training may enhance cognitive function by increasing neurotransmitter levels and blood flow, and improving oxygen supply to the brain (*Babaei & Azari, 2021*). Moreover, cognitive training targeting executive control, working memory, and attention domains can stimulate the release of neurotrophic factors such as BDNF (*Ma et al., 2023*), thereby improving cognitive functions. *Formenti et al. (2020)* found that certain training regimens with high cognitive demands, such as balance training, elicit immediate and positive effects on reaction time, perceptual speed, and executive control, akin to those observed with aerobic exercise. Notably, following immediate mind-body exercises like yoga, participants demonstrated cognitive performance in inhibition and working memory tasks that surpassed even that achieved through aerobic activities (*Gothe et al., 2013*). While several research teams have observed that combining physical exercise with cognitive stimulation may lead to greater enhancements in cognitive performance among older adults compared to physical exercise alone (*Rathore & Lom, 2017*), it is important to note that the effects of various motor-cognitive training modalities on older adults can vary. Thus, further investigation is needed to determine whether different motor-cognitive training modalities produce distinct effects on cognition and brain function.

Finally, previous studies examining long-term interventions involving motor-cognitive training have seldom controlled or documented the intensity of physical activity. Given that the cognitive benefits of exercise are influenced by exercise intensity (*Jeon & Ha, 2017*; *Reycraft et al., 2020*), it is necessary to consider exercise intensity and investigate the impact of various combinations of exercise and cognitive training on cognitive function in older adults. In addition, the effects of long-term exercise interventions accrue through a

series of acute interventions. Therefore, gathering data from acute exercise interventions is essential for predicting the effects of long-term motor-cognitive training.

The present study aims to comprehensively compare and evaluate the immediate effects of two typical motor-cognitive training modalities on cognitive function, postural control, walking ability, and dual-task performance in older adults. It hypothesized that interactive motor-cognitive training (IMCT) will yield superior immediate effects on cognitive functioning in older adults compared to dual-task treadmill training. Additionally, there is a posited temporary decrease in postural control and gait stability among older adults from both acute exercise regimes.

## MATERIALS AND METHODS

### Subjects

*A priori* power analysis was conducted using G*Power, which indicated that with an effect size of 25%, a significant level of 0.05, and a power of 80% power, a total of 21 participants would be needed to detect differences among the conditions over time. Based on the Physical Activity Readiness Questionnaire (PAR-Q), a total of 45 healthy elderly individuals were recruited from two communities in Jinan, Shandong Province, China, using poster advertisements and community-based outreach methods. Participants were eligible to take part if they met the following criteria: (1) aged between 60–75 years; (2) exhibited normal cognitive function, as assessed by the Montreal Cognitive Assessment (MoCA) with a score of ≥26; (3) had no cardiovascular, pulmonary, metabolic, visual, or auditory diseases; (4) had no neurological or psychiatric disorders; (5) were able to walk independently for 6 min; (6) had no contraindications to exercise, as determined through the PAR-Q screening; (7) had not recently taken drugs affecting cognitive or motor performance (*e.g.*, antidepressants and hypnotics).

Out of the 45 subjects initially recruited, four individuals were unable to complete the experiment due to illness or injury, while six subjects voluntarily withdrew their participation, citing a lack of interest. No adverse events were observed during the study. Ultimately, a total of 35 subjects ($M_{age}$ = 66.91 ± 3.15) completed the exercise interventions and all assessment tests (see Table 1). Before the start of the study, all subjects provided written consent. The procedures and methods of the study were ethically reviewed and approved by the Sport Science Ethics Committee of Shandong Sport University to ensure the ethical and safe treatment of human subjects (Number: 2022040).

### Study design

The current study employed a single-blind, balanced crossover experimental design to investigate the acute effects of two distinct motor-cognitive training modes on executive functions, postural control, and single/dual-task gait performance in older adults at a moderate intensity. Treadmill dual-task training (TMDT) and interactive motor-cognition training (IMCT) were selected to represent, respectively, motor-cognitive training with supplementary cognitive tasks and motor-cognitive training with incorporated cognitive tasks, according to Herold's classification of motor-cognitive training (*Herold et al., 2018*). The selection of these exercise modalities was based on a comprehensive consideration of

**Table 1 Characteristics of the subjects (mean ± SD).**

|  | Women ($n$ = 17) | Men ($n$ = 18) | Total ($N$ = 35) |
|---|---|---|---|
| Age (years) | 65.98 ± 2.61 | 67.79 ± 3.43 | 66.91 ± 3.15 |
| Height (cm) | 1.61 ± 0.05 | 1.71 ± 0.04 | 1.66 ± 0.07 |
| Weight (kg) | 66.36 ± 16.27 | 72.04 ± 12.24 | 69.29 ± 14.42 |
| BMI (kg m$^{-2}$) | 25.66 ± 5.78 | 24.54 ± 3.95 | 25.09 ± 4.88 |
| MoCA (point) | 27.25 ± 2.79 | 27.84 ± 1.86 | 27.54 ± 2.35 |

the availability of equipment, safety concerns, and relevant previous research (*Chan et al., 2024*; *Dorfman et al., 2014*). Acute exercise was defined as a single session of exercise lasting between 10 and 60 min (*Themanson & Hillman, 2006*).

The subjects participated in a total of four experimental visits. During the first visit, all subjects provided a written informed consent and completed the questionnaire. Following this, they were familiarized with the experimental protocol, which included a practice trial of two modes of motor-cognitive training, each lasting 10 min, along with procedures for testing executive function, gait, and balance abilities. In the second, third, and fourth visits, each subject completed the three experimental conditions—TDMT, IMCT, and a control condition involving pure reading (RD)—in a random order. Tests assessing executive function, gait, and balance were conducted, along with the collection of blood samples, were conducted before and immediately after the experiments. A minimum of 72 h was required between each experimental session (Fig. 1). Each appointment was scheduled at the same time of day for each participant. Subjects were instructed to refrain from engaging in any other sports activities during the experiment.

The within-group variable was testing occasions, including a pre-test and a post-intervention assessment conducted immediately after the training. The dependent variables included response accuracy (for inhibition function and cognitive flexibility), reaction time (for inhibition function and cognitive flexibility), digit memory span (for working memory), spatiotemporal gait parameters (such as walking speed, step length, stride length, double-support phase time, and gait variability), dual-task cost, cognitive accuracy, and indicators related to the Center of Pressure (COP).

### Interventions

During the IMCT and TMDT experimental conditions, subjects were instructed to begin with a 5-min warm-up exercise, followed by 30 min of motor-cognitive training exercises at a moderate intensity level. Moderate intensity was operationally defined as exercising at "50% to 70%" of heart rate reserve (*Riebe et al., 2015*). Selecting moderate intensity was based on several reasons. Firstly, it is common for individuals, particularly older adults, to engage in moderate or low-intensity exercise regularly, while high-intensity exercise is less prevalent (*Hallal et al., 2012*). Secondly, prioritizing cognitive tasks during dual-task processing has been found to impact subtask performance, leading to reduced walking speed and potential difficulties in sustaining higher intensity exercise, as documented by
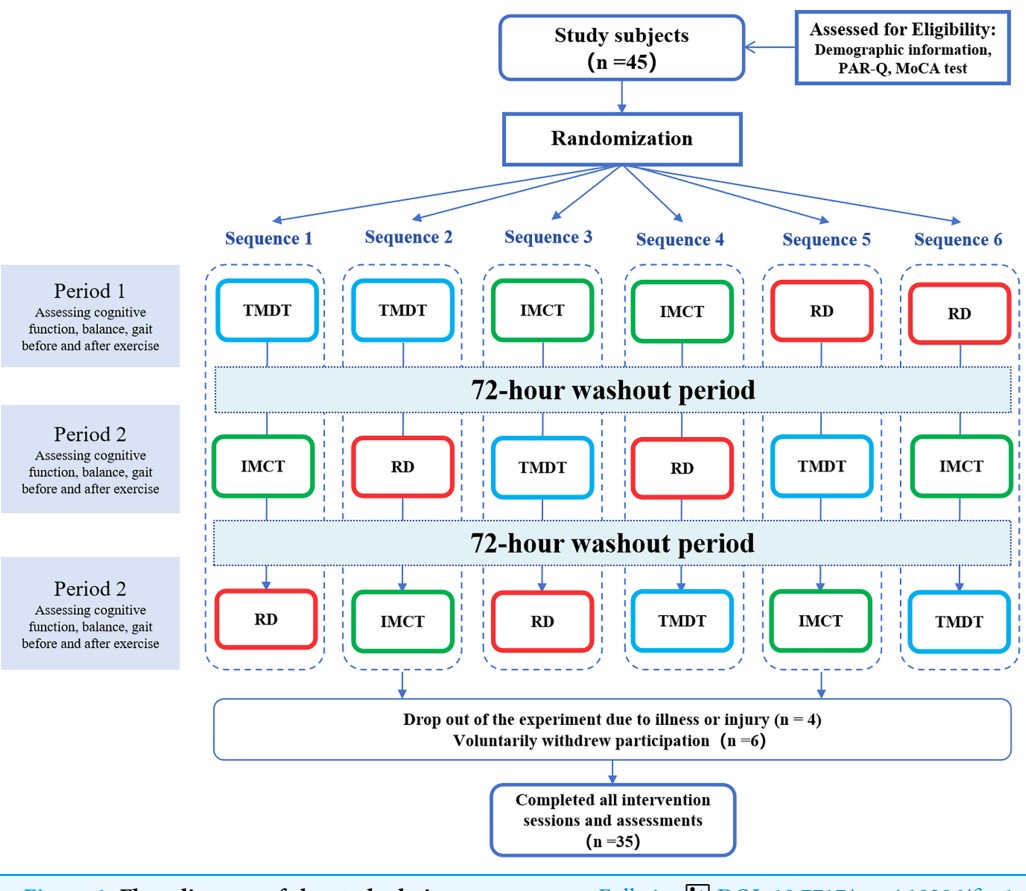

**Figure 1  Flow diagram of the study design.**     

*Yogev-Seligmann et al. (2010)*. Lastly, according to a meta-analysis by *McMorris & Hale (2012)*, moderate-intensity acute aerobic exercise has a significantly greater effect size on cognitive tasks compared to both low-intensity and high-intensity aerobic exercise.

To ensure that subjects maintained a moderate level of intensity during exercise, a combined approach integrating the Rating of Perceived Exertion (RPE) scale with heart rate monitoring was employed. RPE evaluations were conducted every 5 min during the exercise session by one of the investigators. Additionally, each participant wore a Polar M400 heart rate monitor throughout the exercise duration, recording the heart rate at the end of each minute. The target heart rate was determined using the formula developed by *Gellis et al. (2007)* as follows:

Target Heart Rate = $(HR_{max} - HR_{rest}) \times (50\% - 70\%) + HR_{rest}$, where $HR_{max}$ is calculated as $207 - 0.7 \times age$.

### Interactive motor-cognitive training

Utilizing exergames, such as Nintendo Wii Fit and Microsoft Kinect, has been shown to enhance executive function, balance, lower limb strength, gait, and reduce fall risk among healthy older adults (*Jiang, Guo & Wang, 2022*; *Sato et al., 2015*). In this study, the Nintendo Ring Fit Adventure™ (NRFA, Nitendo, Kyoto, Japan) was employed as an

intervention tool for IMCT. The game is a fitness role-playing game that utilizes a Ring-Con connected to a Joy-Con controller. The controller features advanced sensors capable of detecting and translating the player's movements into digital data. The Ring-Con acts as a resistance training tool, enabling users to overcome obstacles or strike objects by either stretching or compressing it. The game can assesse exercise intensity for each player and personalizes adjustments based on their physiological responses. NRFA also offers real-time oral, visual, auditory, and tactile feedback during the game, thereby increasing participants' enjoyment for exercise and enhanc their motor control (*Lamoth, Alingh & Caljouw, 2012*). It not only creates a safe exercise environment but also provides a challenging virtual setting that simulates physical and cognitive tasks that may be difficult to achieve in real life.

NRFA offers various modes, including Adventure, Simple, Custom, and Multitask Modes. For our study, the Simple Mode for subjects was specifically selected due to its emphasis on lower physical skills and gentle movements, such as seated twists, leg raises, and squats, making it more suitable for elderly individuals to participate. Furthermore, a preliminary experiment indicated that the elderly can achieve moderate exercise intensity during this mode.

In the Simple Mode, participants engaged in jogging through various virtual environments and completing various tasks using the Ring-Con accessory. These tasks included jumping, rowing, smashing boxes, and collecting coins. Participants coordinated their actions by squeezing or stretching the fitness ring to carry out these tasks. A total of seven task scenarios were selected, completed sequentially by participants, with each scene requiring 2–7 min for completion.

### Treadmill dual-task training

The treadmill dual-task training (TMDT) protocol requires subjects to engage in cognitive tasks while walking briskly or jogging on a treadmill. A 32-inch high-definition television screen connected to a computer was positioned 0.8 m in front of the treadmill and approximately 1.5 m above the ground, ensuring that the center of the screen aligned with the subjects' eye level. Throughout the exercise session, a researcher operated the computer to present cognitive tasks to the participants. Another researcher, positioned at the side, ensured the safety of the participants and monitored their heart rate. The treadmill's incline was adjusted as needed to maintain target heart rates during the exercise. The cognitive training tasks comprised six different activities, including digital counting, backward counting, reverse recitation, forward and backward recitation, naming, and retelling (refer to Table 2), with each task lasting approximately 3–5 min.

The cognitive training tasks consist of seven different exercises: phoneme monitoring (*Yogev-Seligmann, Hausdorff & Giladi, 2008*), arithmetic tasks (*Dorfman et al., 2014; Tabak, Aquije & Fisher, 2013*), backward recitation, forward and backward recitation, retelling, naming, and backward counting (*Yin, Wang & Liu, 2014*), as outlined in Table 2. Each task lasts approximately 3–5 min. No explicit instructions were given regarding the prioritization of motor and cognitive tasks.

**Table 2 Tasks used for cognitive training.**

| Cognitive tasks | Content |
| --- | --- |
| Phoneme monitoring | Listen to short stories and are asked to answer informative questions at the end of the section. |
| Arithmetic tasks | Calculate simple additions, subtractions, multiplications, and divisions. |
| Reverse recitation | Recite traditional festivals in reverse order, such as naming them backwards |
| Forward and backward recitation | Recite 3 or 4-digit numbers given by the researcher in reverse order (*e.g.*, 9-4-3-7: 7-3-4-9). |
| Retelling | Recite 3 or 4-digit numbers given by the researcher (such as |
| Naming | Name the animals, fruits, plants, or colors that appear alternatively on the screen (*etc.*, horse, strawberry, rose, pink, *etc.*) |
| Counting backwards | Count down 10 numbers from a randomly given number between 50 and 100 provided by the investigator (*e.g.*, given 59, the subject counts backwards: 58, 57, 56, 55, 54, 53, 52, 51, 50, 49). |

### Control session

During the control session, participants were instructed to engage in 30 min of reading magazines while maintaining a quiet state and minimizing physical movement.

## Materials and procedure

### Executive function

Before the tests, subjects were seated comfortably in front of a computer, ensuring their eyes were horizontally aligned and the monitor was positioned 60 cm away. They were instructed to practice and familiarize themselves with the use of each reaction key and the experimental procedure. The familiarization phase took place on the same day as the experimental session, ensuring that participants were well-prepared and comfortable with the tasks they would be undertaking. To further minimize the potential learning effect, participants underwent repeated testing before the formal assessment. Once their accuracy rate in practice reached 80%, indicating they had essentially mastered the entire testing process, the experiment commenced formally.

### Inhibitory function

The modified Stroop task (*Stroop, 1935*) was utilized to assess inhibitory function. Participants were tasked with identifying the color of Chinese characters displayed on a computer screen, which could appear in red, green, yellow, or blue. They were required to respond to the color while inhibiting their semantic response. Each trial began with a fixation point (+) displayed for 1,000 ms, followed by a stimulus presented for 1,500 ms. Participants had to respond within 2,300 ms after the stimulus disappeared. The experiment consisted of one practice block and two formal test blocks. The practice block included 18 trials, while the formal test block comprised 36 randomly presented color-word stimuli. The dependent variables measured were accuracy rate (%) and reaction time (ms).

### Cognitive flexibility

Cognitive flexibility was assessed using the More-Odd Shifting paradigm (*Hillman et al., 2006*). The task involved presenting digits 1–9 (excluding 5) for 2,000 ms each, with a

1,000 ms interval between presentations. Three types of blocks were used in this task. Block A comprised 16 non-switching trials, during which participants determined whether the displayed digit was greater or less than five. Block B also included 16 non-switching trials, but participants were asked to identify whether the displayed digit was odd or even. Block C consisted of 32 switching trials, where participants judged both the size of the digit and the odd/even status of a green digit. A total of six blocks were conducted, maintaining a balanced ABCCBA sequence. The switching cost for each experimental type (switching *vs.* non-switching blocks) was recorded and calculated as the mean reaction time and accuracy for switching and non-switching blocks.

*Working memory*

Working memory was measured using the Digit Span task (*Tripathi et al., 2019*). Participants were presented with a series of digits ranging from 1 to 9 on a screen, each set displayed for 1,000 ms. The length of the digits gradually increased from 2 to 11 digits. After each presentation, participants were instructed to recall the digits in the order they were shown. Three trials were administered for each digit length, and if participants answered correctly on two or more trials, the digit length increased. If not, the test concluded. The maximum length of digits recalled by participants before reaching capacity was recorded as their score on the Digit Span task.

### Postural control

For the evaluation of static balance, K-FORCE Plates (Kinvent, Montpellier, France) was used, which operate at a sampling frequency of 75 Hz. These plates are part of the K-FORCE product series, designed for precise dynamometry in human balance assessment and rehabilitation. Each K-FORCE Plate comprises two platforms, each equipped with two embedded dynamometers. With dimensions of 320 × 160 mm, these plates can capture forces up to 150 kg per plate. The data collected from the plates is transmitted wirelessly *via* a Bluetooth Low Energy (BLE) device connected to the plates. Several studies have shown that the K-FORCE Plate has demonstrated good to excellent internal consistency reliability (0.88 < ICC < 0.93) (*Meras Serrano, Mottet & Caillaud, 2023*) when measuring static balance.

Before and immediately after each training session, participants underwent posturographic trials on K-FORCE Plates, comprising three trials of double-limb stance with eyes open (DLEO) and three trials with eyes closed (DLEC). The assessments were conducted in a quiet environment, following a 5-min period of seated rest for each participant. The force plate was positioned one meter away from a white wall. During the test, participants positioned their feet according to the foot position indicator on the force plate, maintaining an approximate angle of 30 degrees between their feet and a distance of approximately five centimeters between their heels. They kept their arms hanging naturally and directed their gaze toward the wall. Participants were instructed to maintain stability and avoid unnecessary movements while standing still for 30 s. Each subject underwent a practice trial to become acquainted with the task prior to the formal tests. Analysis software (AMTI, BioAnalysis, Version 2.2, Watertown, MA, USA) was used to calculate

various parameters including the center of pressure path length ($COP_L$), average velocity, displacement in the anterior-posterior direction ($COP_{AP}$), and medial-lateral direction ($COP_{ML}$), as well as the enveloping surface during both eyes open and eyes closed stance.

### Gait performance

Gait performance was assessed using an optical measurement system (OptoJump, Microgate, Bolzano, Italy) which comprises photoelectric cells positioned along transmitting and receiving bars. These bars are one meter in length and can extend up to 100 m, with a maximum distance of six meters between them. Infrared LED diodes embedded in the bars facilitate communication between them. The system automatically records spatio-temporal parameters as a subject passes through the bars, triggered by interrupted communication. Data were extracted at 1,000 Hz and saved on a PC using OptoJump Version 1.6.4.0 software (Microgate S.r.I, Bolzano, Italy). The validity and reliability of the OptoJump system for assessing temporal measurements have been established in previous studies. For example, *Hanley & Tucker (2019)* evaluated temporal parameters in racewalkers by utilizing the OptoJump system alongside high-speed video analysis in both overground (0.978 < ICC < 0.988) and treadmill conditions with ICC ranging from 0.890 to 0.988.

During the single-task walking condition, subjects were instructed to complete three trials of walking at a comfortable pace along a 5-m walkway equipped with parallel transmitting-receiving bars. Participants commenced walking from a distance of two meters in front of the walkway and continued until reaching two meters beyond its end. They then returned to the starting point from the outside of the walkway.

In the dual-task walking test, subjects were required to continuously walk back and forth along the walkway for 1 min at a self-selected speed while verbally reciting serial subtractions of three, starting from a randomly assigned number between 200 and 300. The following spatiotemporal parameters were recorded under both gait task conditions: gait velocity (m/s), step length (cm), stride length (cm), double support time (s), double support phase (%), step length coefficient of variation (CV), propulsion phase CV, and double support phase CV (%). Additionally, cognitive task accuracy and dual-task cost (DTC) were calculated. Cognitive task accuracy was determined by dividing the number of correct responses by the total number of attempts (*Howell, Osternig & Chou, 2016*). The calculation formula for DTC is as follows: DTC = (dual-task walking speed − single-task walking speed)/single-task walking speed × 100% (*Chen & Pei, 2018*).

### Blood sampling

A total of 7 mL of venous blood was drawn from the subjects immediately before and after the TMDT and IMCT training sessions. The blood samples were collected in coagulant-free Vacutainer specimen tubes, incubated for 20 min at room temperature to allow clotting, and then centrifuged at 5,000 rpm for 20 min. Following centrifugation, the supernatant serum was carefully collected and transferred into tubes, which were subsequently stored at a temperature of −80 °C. Measurement of BDNF (EK0307; BDNF Human Elisa Kit, Boster, Pleasanton, CA, USA) and dopamine (DA ELISA Kit, E-EL-

0046C; Elabscience, Houston, TX, USA) was performed according to the manual procedures of the human enzyme-linked immunosorbent assay (ELISA) kit.

## Statistical analysis

An independent sample t-test was conducted to determine whether significant differences existed in heart rate and RPE data between the TMDT and IMCT interventions among elderly subjects. The executive function tasks, including the Stroop task, Digit Span task, and More-Odd Shifting task, were designed and administered using E-Prime 3.0 software. Output data were analyzed for accuracy rates and mean response times across the three experimental conditions, excluding data from the practice phase and erroneous responses. Additionally, the data underwent screening for accuracy, outliers, and normality. Further analysis was conducted using SPSS 26.0.

In instances where the data deviated from a normal distribution, a logarithmic transformation was applied. Levene's test was utilized to assess the homogeneity of variances. A mixed-design analysis of variance was employed to explore the interactive effects of experimental conditions and testing occasions on executive functions, postural sway, gait parameters, as well as BDNF and dopamine values, with partial eta squared used as effect size. If a significant interaction effect between time and group be observed, a simple effect analysis was conducted. The significance level ($\alpha$) was set at 0.05.

## RESULTS

### Anthropometric and exercise intensity data

To enhance the validity and reliability of the study findings, the participants' heart rate and RPE were closely monitored during their exercise sessions. This approach aimed to mitigate the influence of exercise intensity on the study outcomes by accounting for potential confounding variables. Independent samples t-tests indicated no significant differences in heart rate and RPE between the TMDT and IMCT conditions (Table 3).

### Executive functions

Table 4 presents the results for the Stroop, More-Odd Shifting, and Digit Span tasks. The Mixed-design ANOVA revealed a significant interaction between treatment and time concerning the correct rate ($F = 4.17$, $p = 0.02$, $\eta_p^2 = 0.076$, see Fig. 2A) and reaction time ($F = 3.47$, $p = 0.04$, $\eta_p^2 = 0.064$, see Fig. 2B) for the Stroop task with incongruent trials. Results from simple effect tests indicate that participants in the IMCT condition displayed a significantly higher correct rate following the training compared to those in the RD condition ($p = 0.014$) and the TMDT condition ($p = 0.042$). However, there was no significant difference between the TMDT and RD conditions ($p > 0.05$) concerning the correct rate. Regarding reaction time in the Stroop color-word incongruent task, participants in both the IMCT condition ($p = 0.041$) and the TMDT condition ($p = 0.044$) exhibited a significant decrease compared to the control condition. However, there was no significant difference between the two types of motor-cognitive training ($p > 0.05$). Additionally, no significant interaction was observed among the three conditions over time for the correct rate and reaction time with the neutral trials ($p > 0.05$).

**Table 3 Comparison of exercise intensity during treadmill dual-task and interactive motor-cognitive interventions.**

|  | TMDT | IMCT | $p$ |
|---|---|---|---|
| Heart rate (bpm) | 114.48 ± 8.42 | 110.92 ± 12.82 | 0.175 |
| RPE | 11.95 ± 1.12 | 12.16 ± 1.11 | 0.421 |

**Table 4 Pre- and posttest changes in executive function ($n$ = 35).**

| Executive function | Task conditions | TMDT | | IMCT | | RD | | Mix-design ANOVA | | |
|---|---|---|---|---|---|---|---|---|---|---|
|  |  | Pre | Post | Pre | Post | Pre | Post | F | $p$ | $\eta_p^2$ |
| Stroop |  |  |  |  |  |  |  |  |  |  |
| Correct rate | Incongruent | 0.84 ± 0.10 | 0.89 ± 0.09 | 0.82 ± 0.08 | 0.93 ± 0.08 | 0.84 ± 0.19 | 0.87 ± 0.13 | 4.17 | 0.02 | 0.08 |
|  | Neutral | 0.95 ± 0.11 | 0.99 ± 0.02 | 0.96 ± 0.07 | 0.97 ± 0.07 | 0.94 ± 0.10 | 0.96 ± 0.10 | 0.44 | 0.65 | 0.01 |
| Reaction time (ms) | Incongruent | 1,284.47 ± 182.19 | 1,091.44 ± 222.32 | 1,296.18 ± 208.76 | 1,089.66 ± 187.31 | 1,262.87 ± 326.37 | 1,198.43 ± 245.67 | 3.47 | 0.04 | 0.06 |
|  | Neutral | 1,014.23 ± 172.06 | 934.72 ± 196.12 | 1,004.36 ± 172.06 | 914.59 ± 145.31 | 1,092.07 ± 226.85 | 997.69 ± 207.62 | 0.06 | 0.94 | 0.01 |
| More-Odd Shifting |  |  |  |  |  |  |  |  |  |  |
| Correct rate | Non-switching | 0.93 ± 0.08 | 0.94 ± 0.07 | 0.94 ± 0.07 | 0.94 ± 0.06 | 0.93 ± 0.08 | 0.95 ± 0.04 | 1.40 | 0.25 | 0.03 |
|  | Switching | 0.77 ± 0.17 | 0.90 ± 0.09 | 0.76 ± 0.17 | 0.89 ± 0.10 | 0.75 ± 0.17 | 0.82 ± 0.12 | 3.15 | 0.47 | 0.06 |
| Reaction time (ms) | Non-switching | 801.54 ± 159.74 | 787.78 ± 156.58 | 708.29 ± 152.51 | 798.95 ± 148.32 | 837.35 ± 160.15 | 814.36 ± 132.54 | 0.68 | 0.51 | 0.01 |
|  | Switching | 1,125.96 ± 187.87 | 995.03 ± 157.42 | 1,135.27 ± 153.27 | 954.99 ± 117.90 | 1,103.61 ± 159.50 | 1,048.93 ± 179.55 | 3.84 | 0.03 | 0.07 |
| Digit Span |  | 7.06 ± 1.49 | 7.11 ± 1.64 | 6.86 ± 1.57 | 7.34 ± 1.68 | 6.63 ± 1.40 | 6.94 ± 1.16 | 1.03 | 0.34 | 0.02 |

In the More-Odd Shifting task, a notable interaction between time and treatment was observed solely in reaction time during the switching tasks ($F$ = 3.84, $p$ = 0.03, $\eta_p^2$ = 0.07, see Fig. 2C). Specifically, the subjects exhibited improved performance in reaction time following the IMCT training compared to those in the RD condition ($p$ = 0.012). However, there was no significant difference in reaction time on the switching task between the TMDT condition and either the IMCT condition ($p$ > 0.05) or the RD condition ($p$ > 0.05) following the acute intervention. In addition, no significant interaction between time and treatment was found for the digit span task ($p$ > 0.05).

## Postural sway

### *Double limb stance with closed eyes*

To address the skewness of the data, a logarithmic transformation was performed to achieve a normal distribution. A mixed-design ANOVA was utilized to compare the effects of the three acute interventions on postural control in older adults. The results showed no significant time × treatment interactions ($p$ > 0.05) for $COP_L$, $COP_{AP}$, $COP_{ML}$, COP velocity, and sway area during the DLCE among the subjects. Moreover, the main effects analysis indicated no significant differences in these variables concerning treatments and time ($p$ > 0.05).

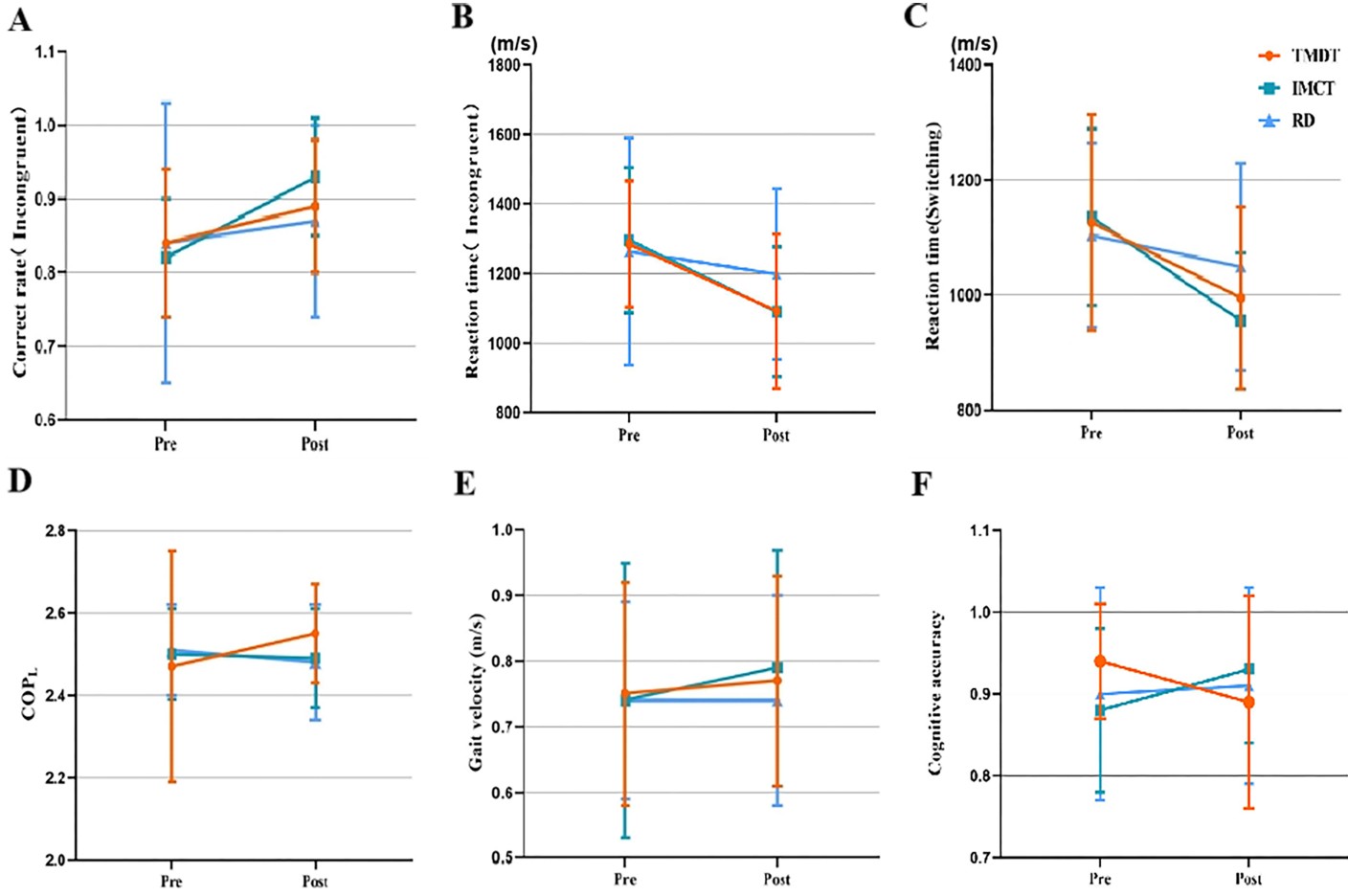

**Figure 2 Interaction effect between time and condition.** (A) Correct rate in incongruent trials of Stroop task. (B) Reaction time in incongruent trials of Stroop task. (C) Reaction time in switching trials of More-Odd Shifting task. (D) COP path length in the 30-s DLOE test. (E) Dual-task velocity. (F) Cognitve accuracy.

### Double limb stance with open eyes

Table 5 presents the changes in COP during the 30-s DLEO test before and after intervention in the TMDT, IMCT, and RD conditions. The results of the mixed-design ANOVA showed a significant time × treatment interaction effect on the COP path length displacement measure in the 30-s DLOE test ($F = 4.54$, $p = 0.013$, $\eta_p^2 = 0.081$; see Fig. 2D). The subsequent simple effects analysis revealed a significant increase in COP path length under the DLOE test condition following acute TMDT training in older adults ($F = 8.04$, $p = 0.006$, $\eta_p^2 = 0.072$). Although significant differences among the three interventions were evident in the post-test ($F = 3.154$, $p = 0.047$, $\eta_p^2 = 0.058$), no significant differences existed among the three interventions at baseline ($F = 0.412$, $p = 0.663$, $\eta_p^2 = 0.008$).

No significant interaction effects were found in sway area, sway velocity, $COP_{AP}$, and $COP_{ML}$ ($p > 0.05$). The main effects analysis revealed no significant differences in these measures concerning both time and group effects ($p > 0.05$).

**Table 5 Pre- and posttest changes in COP during the 30-s DLEO test ($n$ = 35).**

| COP | TMDT | | IMCT | | RD | | Mix-design ANOVA | | |
|---|---|---|---|---|---|---|---|---|---|
| | Pre | Post | Pre | Post | Pre | Post | F | $p$ | $\eta_p^2$ |
| Sway area | 2.07 ± 0.29 | 2.10 ± 0.26 | 2.07 ± 0.30 | 2.04 ± 0.33 | 2.04 ± 0.26 | 2.03 ± 0.31 | 0.35 | 0.71 | 0.007 |
| $COP_L$ | 2.47 ± 0.28 | 2.55 ± 0.12 | 2.50 ± 0.11 | 2.49 ± 0.12 | 2.51 ± 0.11 | 2.48 ± 0.14 | 4.54 | 0.01 | 0.081 |
| COP velocity | 1.04 ± 0.13 | 1.07 ± 0.12 | 1.02 ± 0.11 | 1.01 ± 0.12 | 1.03 ± 0.11 | 1.01 ± 0.13 | 2.31 | 0.10 | 0.043 |
| $COP_{AP}$ | 1.16 ± 0.16 | 1.16 ± 0.14 | 1.12 ± 0.15 | 1.15 ± 0.17 | 1.15 ± 0.13 | 1.12 ± 0.16 | 0.98 | 0.38 | 0.019 |
| $COP_{ML}$ | 1.02 ± 0.17 | 1.04 ± 0.17 | 1.05 ± 0.18 | 1.00 ± 0.20 | 1.00 ± 0.16 | 1.01 ± 0.19 | 1.75 | 0.18 | 0.033 |

## Gait performance

No significant time × treatment interactions were observed for any spatio-temporal gait parameters for the single-task walking test ($0.15 < p < 0.87$, $0.00 < \eta_p^2 < 0.04$). However, a significant time × treatment interaction was found for gait velocity during the dual-task walking test ($F = 3.626$, $p = 0.03$, $\eta_p^2 = 0.068$; Fig. 2E). The results of the simple effect test indicated a significant increase in gait velocity among subjects following acute IMCT training (0.79 ± 0.18 m/s) compared to their pre-test performance (0.74 ± 0.21 m/s, $p < 0.001$). No significant difference was found in gait velocity for the TMDT and RD conditions before and after intervention ($p > 0.05$). Additionally, there were no significant differences in gait velocity among the different intervention groups at both baseline and post-test levels ($p > 0.05$). The detailed mix-design ANOVA results including means and differences are provided in Table 6.

There was a significant interaction effect of time and treatment on cognitive accuracy ($F = 3.427$, $p = 0.036$, $\eta_p^2 = 0.065$; Fig. 2F). However, the results of the simple effect test did not reveal any discernible differences between the interventions or testing occasions. This could be attributed to either a small sample size or high variability among subjects. Notably, cognitive accuracy showed a trend toward improvement after IMCT ($p = 0.07$), while it exhibited a notable decline after TMDT ($p = 0.06$), although this decrease did not reach statistical significance.

No significant interaction effect between time and treatment was found for dual-task cost ($p > 0.05$). However, a significant main effect was observed in dual-task cost for time ($p = 0.042$). The consistent performance of the reading group before and after the intervention (see Table 6) implies that the observed differences in pre- and post-intervention can largely be attributed to the effects of the two types of motor-cognitive training.

## BDNF and dopamine

A total of 15 subjects completed the collection of all four blood samples. No significant time × treatment interaction was found for BDNF values ($p > 0.05$). Although there was no significant time × treatment interaction in terms of dopamine, the $p$-value was close to

**Table 6 Pre- and posttest changes in gait performance during the dual-task walking test (*n* = 35).**

| Spatio-temporal gait parameters | TMDT | | IMCT | | RD | | Mix-design ANOVA | | |
|---|---|---|---|---|---|---|---|---|---|
| | Pre | Post | Pre | Post | Pre | Post | *F* | *p* | $\eta_P^2$ |
| Gait velocity (m/s) | 0.75 ± 0.17 | 0.77 ± 0.16 | 0.74 ± 0.21 | 0.79 ± 0.18 | 0.74 ± 0.15 | 0.74 ± 0.16 | 3.63 | 0.03 | 0.07 |
| Step length (cm) | 52.29 ± 5.31 | 52.42 ± 6.11 | 52.56 ± 7.11 | 52.00 ± 6.34 | 51.14 ± 5.37 | 51.65 ± 6.19 | 1.14 | 0.32 | 0.02 |
| Stride length (cm) | 103.55 ± 11.16 | 104.08 ± 12.55 | 103.81 ± 13.51 | 103.48 ± 13.17 | 101.94 ± 11.74 | 102.02 ± 12.45 | 0.22 | 0.81 | 0.004 |
| Double support time (s) | 0.43 ± 0.14 | 0.42 ± 0.10 | 0.46 ± 0.26 | 0.44 ± 0.16 | 0.43 ± 0.11 | 0.42 ± 0.14 | 0.04 | 0.97 | 0.001 |
| Double support phase (%) | 58.79 ± 6.43 | 55.29 ± 13.41 | 59.39 ± 4.56 | 58.64 ± 6.32 | 59.79 ± 7.57 | 59.59 ± 5.99 | 1.57 | 0.21 | 0.03 |
| Step length CV (%) | 21.8 ± 8.21 | 21.44 ± 5.58 | 26.42 ± 13.06 | 22.19 ± 11.16 | 22.68 ± 7.15 | 23.24 ± 7.3 | 2.36 | 0.10 | 0.05 |
| Propulsion phase CV (%) | 89.45 ± 31.78 | 87.07 ± 32.88 | 91.81 ± 36.12 | 88.79 ± 34.49 | 94.57 ± 31.00 | 85.26 ± 28.72 | 0.46 | 0.63 | 0.01 |
| Double support CV (%) | 43.4 ± 28.64 | 38.53 ± 20.68 | 39.63 ± 24.69 | 42.53 ± 21.54 | 41.73 ± 32.46 | 41.03 ± 35.12 | 0.47 | 0.63 | 0.01 |
| Dual-task cost | 0.35 ± 0.13 | 0.33 ± 0.11 | 0.35 ± 0.15 | 0.32 ± 0.12 | 0.34 ± 0.11 | 0.34 ± 0.10 | 1.37 | 0.26 | 0.03 |
| Cognitive accuracy | 0.94 ± 0.70 | 0.89 ± 0.13 | 0.88 ± 0.10 | 0.93 ± 0.09 | 0.90 ± 0.13 | 0.91 ± 0.12 | 3.43 | 0.04 | 0.07 |

reaching significance, with the increase in mean value in the IMCT condition (0.126) much greater than that in the TMDT group (0.048) before and after the experiment.

## DISCUSSION

The present study, employing a randomized controlled crossover design, aimed to investigate the effects of moderate-intensity TMDT and IMCT on cognitive functions, postural control, and gait stability in older adults. A significant improvement in executive functions among participants was observed for both motor-cognitive training modalities. Interestingly, IMCT demonstrated superior outcomes across several domains of executive functions compared to TMDT. Notably, TMDT might adversely affect postural control in older adults when their eyes are open, whereas IMCT immediately enhanced walking speed, reduced dual-task cost, and improved cognitive accuracy during dual-task performance in this population. This study addresses a gap in the existing literature by comparing the effects of different motor-cognitive training modalities on the physical and mental health of the elderly, which is crucial for promoting their overall wellbeing and mitigating the risk of chronic diseases (*Jo et al., 2020*), as emphasized by previous research (*Cugusi, Prosperini & Mura, 2021*).

### The effect of motor-cognitive training on the executive function of the elderly

#### Inhibitory function

In the Stroop paradigm, subjects are tasked with responding to a specific stimulus dimension while inhibiting another competing stimulus dimension. The findings of this study suggest that when the meaning of the word conflicts with the color, the IMCT group demonstrated greater accuracy in task processing compared to both the TMDT and RD groups. However, no significant difference in accuracy was observed between the TMDT and RD groups. Moreover, the reaction times of both motor-cognitive training groups

were quicker than those of the reading group. These results are consistent with previous studies conducted over both short-term and long-term periods.

The impact of acute physical exercise on inhibitory function has been examined in several studies (*Netz et al., 2023*). The findings from *Chang et al. (2019)* demonstrate that irrespective of the duration of aerobic exercise (20 or 45 min), participants exhibited shorter reaction times in the Stroop test compared to the control group. A meta-analysis further corroborated these findings, suggesting that participating in moderate or high-intensity physical exercise for over 10 min can notably improve inhibitory function in older adults (*Cai et al., 2020*).

The systematic review and meta-analysis conducted by *Jiang, Guo & Wang (2022)* demonstrated a significant positive impact of long-term interactive motor-cognitive intervention on three aspects of cognitive function in older adults: inhibition, updating, and switching. Notably, the intervention showed the most pronounced effect in improving inhibitory function, which is known to be highly responsive to exergame intervention (*Dhir et al., 2021*). This phenomenon can be easily explained. Firstly, IMCT, like exergames and virtual reality games, demands constant restraint or inhibition of inappropriate responses. For example, games such as NRFA require players to respond solely to rewards like gold coins while suppressing responses to distracting elements like 'bombs'. Hence, such training serves as a valuable exercise in enhancing inhibition ability. Secondly, research suggests that open motor skills have a greater impact on inhibitory control compared to closed motor skills (*Liu et al., 2020*). In the present study, treadmill dual-task training represents a closed action skill, involving fixed sports venues, equipment, and movement patterns. In contrast, IMCT involves a dynamic virtual environment, requiring the processing of changing information. The ability to adapt to changes, forecast future events, and make judgments calls are characteristics of open motor skills. These distinctions may contribute to the differential effects on inhibitory function between the two types of motor-cognitive training.

When the color of the word aligned with its meaning in the neutral condition, there was no notable alteration in reaction time and accuracy across groups and time points. This can be ascribed to the overall health and normal cognitive function of the elderly participants involved in the study. The neutral tasks may not have exerted cognitive pressure or posed a challenge to these individuals, thereby potentially resulting in a ceiling effect. This is evidenced by the accuracy rates for the neutral condition in all groups, which surpassed 95%, as illustrated in Table 4.

### Cognitive flexibility

Cognitive flexibility refers to the ability to switch between thinking about two different concepts or thinking about multiple concepts simultaneously (*Dajani & Uddin, 2015*). In humans, aging leads to a natural decline in cognitive flexibility (*Brauer Boone, Miller & Lesser, 1993*). Notably, mild cognitive impairment (MCI) represents a stage that lies between the typical memory changes associated with aging and the onset of Alzheimer's disease. Individuals diagnosed with MCI typically experience a further decline in cognitive flexibility (*Jiménez, Arellano & Avilés, 2013*). Therefore, preserving cognitive flexibility in

older adults may serve as a potential strategy to mitigate the adverse effects of aging on cognitive decline.

The results of the current study indicated that older adults engaged in IMCT training demonstrated significantly reduced reaction times in the More-Odd Shifting switching task compared to the control group. However, there was no discernible difference in reaction times between the IMCT and TMDT training groups. These findings suggest that IMCT might offer potential advantages in enhancing older adults' capacity to handle mixed tasks. Numerous studies have also highlighted the beneficial effects of IMCT on cognitive flexibility, encompassing switching and updating abilities. This may be attributed to the diverse game environments in IMCT, which require subjects to switch between different tasks and rules and update their existing memory resources to swiftly adapt to changing circumstances and optimize game performance (*Adcock et al., 2020*; *Gschwind et al., 2015*).

In the non-switching task condition of More-Odd Shifting, there were no significant differences in response time and accuracy among older adults before and after the three interventions. This observation may be ascribed to the cognitive health status of the study participants, who were all older adults with no reported cognitive impairments. Moreover, the non-switching task likely did not impose significant cognitive demands or challenges on this demographic, leading to a ceiling effect and making it difficult to observe differences between post-tests and between groups.

### Working memory

Working memory, a fundamental component of cognitive function, involves the temporary storage and processing of information during cognitive tasks (*Baddeley, 1992*). However, this cognitive ability tends to decline with age, particularly after the age of 60 (*Elliott et al., 2011*). Studies have indicated that age-related dual-task performance deficits (*Dubost et al., 2006*; *Granacher et al., 2011*; *Göthe, Oberauer & Kliegl, 2007*) and corresponding changes in neural activation (*Chmielewski, Yildiz & Beste, 2014*) may stem from underlying working memory dysfunction (*Gazzaley & Nobre, 2011*). Physical exercise is considered a safe and effective alternative to delay working memory decline (*Mcsween et al., 2019*).

Some researchers have observed immediate enhancements in working memory following a single session of exercise. For instance, *O'Brien et al. (2017)* reported subjects who engaged in 60–80 min of open-skilled exercises (such as aerobic exercise, tennis, or dance) or closed-skilled exercises (such as fitness or swimming) demonstrated significant improvements in digit span performance compared to active control groups. *Tsujii, Komatsu & Sakatani (2013)* reported that after 10 min of moderate-intensity cycling (at 40% $VO_2max$), healthy older adults exhibited significantly lower reaction times in digit span compared to a blank control condition. Moreover, long-term studies suggest that incorporating cognitive tasks into exercise training, thereby combining the benefits of cognitive and physical exercises, can yield even greater cognitive improvements (*Moreau, 2015*; *Paas & Sweller, 2012*; *Ruiter, Loyens & Paas, 2015*).

However, this study did not observe the immediate effects of two types of motor cognitive training on working memory in older adults. *Kelly et al. (2014)* suggested that targeted cognitive training may enhance specific cognitive domains, whereas general cognitive training might not effectively engage specific cognitive functions. This aligns with the findings of a meta-analysis by *Wollesen et al. (2020)*, which indicated that motor-cognitive training, including sensorimotor games, significantly enhances inhibitory control in older adults but does not have a significant effect on working memory. Moreover, this study was an acute experimental investigation. However, it is important to note that a single session of training may not be adequate to yield significant differences between the two conditions.

Furthermore, the lack of sensitivity in working memory assessment tools may contribute to the lack of significant differences in the effects of two types of motor-cognitive training. Meta-analysis results by *Cai et al. (2021)* indicate that the intervention effect of exercise on working memory in older adults is influenced by measurement tools: the digit span backward effect size is the largest (SMD = 0.92), spatial span is a medium effect size (SMD = 0.77), and the rest exhibit small effect sizes (including digit span tests). Therefore, future research should employ more accurate working memory measurement tools, such as the digit-letter sequencing task (*Wells et al., 2018*), to further investigate the effects of motor-cognitive training on working memory.

The study results reveal that neither motor-cognitive training significantly affected postural control in healthy elderly individuals with closed eyes. However, it was observed that after dual-task training on the treadmill, there was a significant increase in sway trajectories among the elderly when their eyes were open. These findings partially corroborate previous studies, suggesting that exercise-induced fatigue can have immediate effects on postural control.

## The effect of motor-cognitive training on postural control in the elderly

The findings of this study indicate that neither motor-cognitive training method had a significant impact on postural control in healthy elderly individuals under the eyes closed condition. However, following TDMT, a notable increase in COP path length was observed during the DLEO test, indicating an immediate effect on postural control in older adults (*Bizid et al., 2009*; *Egerton, Brauer & Cresswell, 2009*).

The literature suggests that physical activities engaging major muscle groups in the sagittal plane, like cycling or running, may result in changes in center of pressure (COP)-related outcomes. For instance, *Stemplewski et al. (2012)* investigated the impact of acute moderate-intensity cycle ergometer training on postural control in males aged 65 to 74. Their study revealed an elevation in COP swaying velocity when participants performed the task with their eyes open. Similarly, *Donath et al. (2013)* conducted a maximal exhaustive treadmill exercise test on 19 healthy older adults and noted a notable increase in COP path length. In the current investigation, elderly participants predominantly mobilized their lower limbs in sagittal plane motions during the TDMT intervention, which might lead to an augmented COP sway trajectory. This temporary alteration in COP

path length could potentially elevate the risk of falls among the elderly following exercise (*Piirtola & Era, 2006*).

The longer sway trajectories observed in older adults during the eyes-open state may indicate an increased dependence on visual feedback following acute exercise. Typically, the visual system plays a pivotal role in furnishing spatial orientation and balancing cues. However, the temporary sensory and neuromuscular disruptions experienced by older adults after acute exercise may lead to an increased dependency on visual feedback for maintaining balance. Consequently, they may be more susceptible to balance impairment in situations where visual information is inadequate. Additionally, older adults often encounter temporary challenges in regulating balance immediately post-exercise. Thus, it is imperative to allow an adequate period of rest following high-intensity exercise to facilitate the restoration of neuromuscular system function.

It is worth noting that although the length of the $COP_L$ with eyes open increased in older adults after TMDT, the results of this study indicate that IMCT with the same exercise intensity did not lead to any changes in COP-related outcomes. Aging is commonly linked with a heightened dependence on visual feedback during tasks involving standing balance (*Colledge et al., 1994*). Given the crucial role of vision in the feedforward mechanism of balance, *Chapman & Hollands (2006)* suggest that particular emphasis should be placed on integrating visual feedback into dynamic balance training tasks. Numerous studies have demonstrated that training with visual feedback is more effective than traditional training in enhancing balance function. For instance, *Sihvonen, Sipilä & Era (2004)* discovered that a 4-week visual feedback-based balance training significantly improved the balance ability of frail older adults. In IMCT training, participants engage in various activities, such as climbing steps, navigating obstacles, and targeting objects rapidly, according to specific instructions and time constraints. The game provides immediate visual and auditory feedback based on the player's actions, enabling older adults to promptly discern changes in their posture information, thereby enhancing their balance abilities.

On the other hand, compared to the TMDT, the IMCT used in this study involves a range of movements, including squats and weight shifting, which demand heightened attention to posture control. *Drozdova-Statkevičienė et al. (2021)* found improvements in postural control among older males following acute strength training intervention. The authors proposed that this improvement could stem from the heightened attentional control of posture after exercise (*Woollacott & Cook, 2002*), along with elevated levels of testosterone and cortisol, which serve as mediating factors for improved attentional control and balance (*Drozdova-Statkevičienė et al., 2021*). While the evidence suggests that the IMCT employed in this study may offer greater safety and potential benefits for posture control in older adults compared to TMDT, it is important to acknowledge that the impact of different motor-cognitive training methods on posture control only shows a moderate effect size. Moreover, without additional neurophysiological measurements, the potential mechanisms driving the positive effects of acute motor-cognitive training on posture control cannot be directly assessed based solely on the current research findings.

## The effect of motor-cognitive training on gait performance in elderly adults

The findings of this study suggest that there were no discernible changes in the single-task gait performance of the elderly following both TMDT and IMCT training. Previous research exploring the impact of walking-based exercises on gait parameters has presented varied outcomes, possibly due to differences in exercise duration and intensity. For instance, *Morrison et al. (2016)* conducted a study involving individuals across various age brackets (30–39, 40–49, 50–59, 60–69, and 70–79 years) who were asked to walk uphill on a treadmill until they experienced fatigue. Their findings indicated significant enhancements in gait speed, cadence, and stride length across all age groups. The researchers attributed these changes to a temporary carryover effect of fast walking, whereby subjects maintained a faster walking speed during the subsequent gait test. However, this finding raises concerns regarding older adults, as the immediate increase in walking speed was accompanied by a decrease in various metrics related to balance ability, indicating reduced stability. This assertion is supported by other studies showing that older adults (aged 60–69 and 70–79) with higher walking speed and cadence face an elevated risk of falls (*Morrison et al., 2016*; *Nagano et al., 2014*; *Pereira & Gonçalves, 2011*).

Other researchers have found that acute walking-based exercise does not significantly affect the single-task gait performance of older adults. Even when changes in gait parameters occur, they do not substantially affect the risk of falls in the elderly. For instance, *Donath et al. (2014)* discovered that the spatiotemporal characteristics of gait in older adults did not show significant changes following a maximum incremental exercise test and a 2 km walk. Similarly, *Kirkwood et al. (2011)* conducted a 20-min treadmill walking training session with elderly women and found that although the exercise induced fatigue, it was insufficient to increase the risk of falls among them.

The results of this study align with the findings of *Donath et al. (2014)* and *Kirkwood et al. (2011)*. However, it is essential to highlight that, unlike previous experiments, this study employed motor-cognitive training as an intervention method rather than simple treadmill exercise. The Central Benefit Model of *Liu-Ambrose et al. (2013)* suggests that executive function may influence the risk of falls through various pathways, including changes in attention(*Dault, Frank & Allard, 2001*), gait (*Holtzer et al., 2006*), central processing and integration, and execution of postural responses (*Caetano et al., 2019*). Neuroimaging studies further reinforce this hypothesis, indicating that the functionality of brain regions linked with selective attention and conflict resolution correlates independently with alterations in fall risk profiles (*Liu-Ambrose et al., 2013*). Considering the positive effect of the motor-cognitive dual task on executive functions observed in this study, cognitive stimulation may have a beneficial impact on attention concerning postural control, mitigating the repercussions of motor or movement-related fatigue on gait stability.

Dual-task gait performance showed significant improvement following IMCT, while no significant changes were observed before and after TMDT and pure reading. Specifically, gait speed during dual-task significantly increased after the acute IMCT, with a moderate

effect size. Gait speed serves as a crucial indicator of the overall health status of elderly individuals, with slower walking speeds correlating with heightened susceptibility to clinical disorders (*van Kan et al., 2009*). For each 0.1 ms$^{-1}$ decrease in self-determined walking speed, there is a 10% reduction in independent daily activities among the elderly (*Judge, Õunpuu & Davis, 1996*). This decline in physical activity level further contributes to a deterioration in physical function and an increased risk of falls. Although there is currently a lack of comparable acute intervention data, the findings of this study align with previous long-term motor-cognitive intervention experiments.

*Schättin et al. (2016)* investigated the effects of IMCT and balance training on spatiotemporal gait parameters (walking speed, cadence, and stride length) in healthy older adults, both in single and dual-task conditions. The results indicated that post-intervention, the IMCT group exhibited significant enhancements in walking speed and cadence during dual-task self-paced walking, as well as in walking speed and stride length during dual-task fast-paced walking. Conversely, the balance training group primarily demonstrated improvements in single-task gait parameters. These findings suggest that IMCT may effectively mitigate dual-task interference, improve attention allocation, and consequently decrease fall risks. In another study, *Liu et al. (2022)* conducted a 12-week intervention comparing traditional Tai Chi with interactive Tai Chi in older adults with mild cognitive impairment. They found that compared to the control group, both motor-cognitive training interventions resulted in improvements in dual-task walking speed and cadence in older adults, with no significant difference between the two approaches. Additionally, various systematic reviews have assessed the impacts of motor-cognitive interventions on dual-task performance in healthy older adults (*Maayan et al., 2014*; *Plummer et al., 2015*; *Wollesen et al., 2015*). These studies indicated that compared to single-task training, motor-cognitive dual-task interventions can improve postural control and gait performance under dual-task conditions.

According to previous studies (*Biganzoli et al., 2013*; *Bloem et al., 2001*; *Yogev-Seligmann, Hausdorff & Giladi, 2008*), cognitively healthy older adults typically prioritize gait and posture over cognitive performance during challenging dual-task tests. It has been observed that gait parameters change as cognitive load increases. The findings of the present study suggest that the enhancements in gait resulting from IMCT were not attributable to a shift in priority from cognitive to motor functions, as cognitive accuracy also concurrently improved among the subjects. This implies an improved ability to allocate cognitive resources without significantly compromising gait.

## The effect of motor-cognitive training on dual-task cost and cognitve task accuracy

Dual-task cost refers to the decrease in performance or speed when individuals attempt to simultaneously execute two tasks, as opposed to performing each task individually (*Plummer-D'Amato et al., 2008*). This phenomenon highlights the competition for attention resources that individuals encounter when multitasking. Studies have shown that when the dual-task cost exceeds 20%, it can lead to an unstable gait and an increased risk of falling among individuals (*Hollman et al., 2007*). Moreover, research has linked a dual-task

cost exceeding 20% to the onset of dementia in elderly individuals with moderate cognitive impairment (*Darweesh, Verlinden & Ikram, 2017*). In the current study, the dual-task cost of the subjects showed no changes before (M = 0.34 ± 0.11) and after reading (M = 0.34 ± 0.10). However, there was a noticeable decrease in the dual-task cost by 6.1% and 9.4% after TMCT and IMCT training, respectively. Therefore, these variations likely originate from the effects of the two motor-cognitive training modalities.

One notable difference between TMDT and IMCT training is that while the reduction of dual-task costs in the elderly is evident after IMCT, it is also accompanied by a significant increase in dual-task gait speed, as previously mentioned. Previous research has also noted improvements in dual-task performance among community-dwelling older adults who underwent IMCT. For instance, *Wang et al. (2021)* conducted a study involving 20 community-dwelling elderly individuals, who received IMCT three times a week for 12 weeks. The researchers found that the group undergoing IMCT showed substantial enhancements in executive function and dual-task performance, compared to the control group. Furthermore, the overall enhancement in executive function was strongly correlated with cognitive dual-task performance (r = −0.701). Likewise, *Liu et al. (2022)* compared the effects of traditional Tai Chi and exergaming-based Tai Chi on cognitive function and dual-task gait performance of elderly individuals with mild cognitive impairment over a 12-week period. The results revealed that the dual-task cost decreased by 17% in the traditional Tai Chi group and 22% in the exergame-based Tai Chi group. These findings suggest that IMCT plays a superior role in enhancing the elderly's ability to allocate attention and process information.

Although this study employed an acute intervention method, given the significant effect size reported in previous studies on motor-cognitive training for older adults and the data observed in the current study, it can be confidently asserted that a single session of IMCT enhances cognitive capacity in older adults. This leads to a temporary increase in information processing speed and a reduction in interference between cognitive tasks and walking tasks (*Redfern et al., 2002*). These immediate benefits are also likely to result in long-term improvements in both gait and cognition if the IMCT is administered over several weeks (*Anderson-Hanley et al., 2012*).

Regarding cognitive accuracy, although the interaction between time and intervention method was found to be significant, subsequent analysis of simple effects did not reveal significant results. This could be attributed to the higher variability within groups, possibly necessitating a larger sample size to detect statistically significant differences between them. However, it is worth noting that there was an improvement in cognitive accuracy after IMCT compared to before training, which was approaching significance ($p = 0.07$). On the other hand, the cognitive accuracy following treadmill dual-task training showed a greater decrease in amplitude, with the decrease being close to the significant level ($p = 0.06$). This could be attributed to the longer duration and higher density of cognitive stimulation during treadmill dual-task training, possibly resulting in mental fatigue among participants. Mental fatigue is a result of prolonged periods of performing demanding, cognitive-load-inducing activities, and it reduces efficiency in cognitive performance (*Tanaka, Ishii & Watanabe, 2014*). Thus, integrating suitable cognitive challenges within

motor-cognitive training regimens could hold promise for enhancing cognitive accuracy in older adults.

Given the current lack of validated tools for assessing cognitive load during motor-cognitive dual-task scenarios, measurement of this metric was refrained from. Future research endeavors could concentrate on refining quantitative measures to investigate cognitive load in motor-cognitive training. This would lay a solid foundation for evidence-based motor-cognitive training initiatives.

### BDNF and dopamine

BDNF is primarily expressed in the hippocampus, cortex, and skeletal muscle. Its presence in the brain allows it to traverse the blood-brain barrier into the peripheral blood (*Pan et al., 1998*), and its serum levels correlate positively with those in the brain (*Klein et al., 2012*), indicating that peripheral BDNF levels may reflect changes in brain BDNF levels. Studies have shown that BDNF mechanistically mediates exercise-induced proliferation of hippocampal dentate gyrus cells and is necessary for exercise-induced cognitive enhancement, such as memory and learning. Beyond its cognitive functions, BDNF is intricately involved in various neural processes such as neuronal differentiation, plasticity, cell survival, and hippocampal function. Numerous studies support the notion that BDNF plays a predominant role in regulating the effects of physical activity on cognitive changes. In the present investigation, both types of motor-cognitive training notably elevated circulating levels of BDNF, with no notable distinction between the two groups. Acute exercise and aerobic exercise training increased circulating BDNF concentrations, but resistance training did not (*Dinoff et al., 2017*; *Dinoff et al., 2016*). Furthermore, research has shown that vigorous exercise leads to an increase in BDNF release within the human brain, indicating that exercise mediates the production of central BDNF in humans, with factors such as exercise type, duration, and intensity playing crucial roles. In the current study, both groups participated in aerobic exercise training, with controlled intensity and duration, resulting in comparable BDNF levels between the two groups.

Dopamine is an important catecholamine neurotransmitter in the central nervous system of mammals, serving as a precursor to norepinephrine. It operates in the brain *via* corresponding receptors (*Apitz & Bunzeck, 2013*). Dopamine (DA) plays a vital role in regulating cognitive-related synaptic plasticity, which has an inverted U-shaped relationship with DA levels (*Thirugnanasambandam et al., 2011*). Animal studies have highlighted the connection between dopamine and motivation and reward circuits (*Hu, 2016*), while human genetic research has linked higher dopamine levels in the frontal lobe to sustained attention (*Aalto et al., 2005*) and working memory capacity (*Vijayraghavan et al., 2007*). This relationship suggests that as brain dopamine concentration rises, so does working memory performance. A positive correlation between dopamine expression levels and cognitive task scores was similarly found in the research. Furthermore, the average increase in dopamine levels under IMCT conditions (0.126) was significantly higher than under TMDT (0.048). The impact of different motor-cognitive training on cognitive function is mainly reflected in the technical characteristics of the movement itself and the sports context.

## LIMITATIONS

The findings of the present study shed light on the impact of motor-cognitive training on cognitive and physical functions in older adults. However, limitations such as a small sample size may constrain the generalizability of results. To enhance future research, larger and more diverse samples are recommended. Additionally, the specific motor-cognitive training protocols used in this study (IMCT and TMDT) may not fully represent all types of motor-cognitive interventions, necessitating further exploration of effectiveness of various training methods. While the study compared acute effects, longitudinal research is crucial for understanding sustained impacts on cognitive performance and physical functions over time. Longitudinal studies can provide a more definitive assessment of the causal effects of motor-cognitive training on cognitive and physical functions, as well as neurotransmitter regulation.

## CONCLUSIONS

In conclusion, the study highlights the immediate positive impact of IMCT on cognitive processing, postural control, and dual-task performance in older adults. The results suggest that incorporating elements of IMCT as demonstrated in the study, can be a strategic approach to enhancing cognitive and physical functions among the elderly population. These findings have significant practical implications for practitioners working with older individuals. Health professionals and caregivers can integrate IMCT into regular physical activity programs for older adults to improve overall cognitive and physical health. Moreover, the absence of significant adverse effects on postural control further supports the, adoption of IMCT over TMDT. This evidence can guide policy-makers and practitioners in developing and promoting tailored interventions that address the multifaceted challenges of aging. Incorporating IMCT within existing health and fitness programs could lead to improved quality of life, independence, and reduced healthcare costs associated with falls and cognitive decline in the elderly population.

## ACKNOWLEDGEMENTS

We extend our sincerest gratitude to all the participants who generously gave their time and effort to take part in this study. Without their commitment, this research would not have been possible. We also express our appreciation to the community workers for their invaluable assistance in recruitment and data collection. Additionally, we would like to thank our families and friends for their unwavering support and encouragement throughout this endeavor.

### Funding

This work was funded by the Shanghai Pujiang Program (22PJC094). The funders had no role in study design, data collection and analysis, decision to publish, or preparation of the manuscript.

## Grant Disclosures

The following grant information was disclosed by the authors:

Shanghai Pujiang Program: 22PJC094.

## Competing Interests

The authors declare that they have no competing interests.

## Author Contributions

- Ran Li conceived and designed the experiments, performed the experiments, analyzed the data, authored or reviewed drafts of the article, and approved the final draft.
- Ping Qu conceived and designed the experiments, analyzed the data, authored or reviewed drafts of the article, and approved the final draft.
- Xue Hu conceived and designed the experiments, authored or reviewed drafts of the article, and approved the final draft.
- Xiaojing Li performed the experiments, prepared figures and/or tables, and approved the final draft.
- Haiqing Zeng performed the experiments, prepared figures and/or tables, and approved the final draft.
- Binghong Gao conceived and designed the experiments, authored or reviewed drafts of the article, and approved the final draft.
- Zhiyuan Sun performed the experiments, analyzed the data, authored or reviewed drafts of the article, and approved the final draft.

## Ethics

The following information was supplied relating to ethical approvals (*i.e.*, approving body and any reference numbers):

Shandong Sport University granted ethical approval to carry out the study with its facilities (Ethical Application Ref: 2022040).

## Data Availability

The raw measurements are available in the Supplemental Files.

## Supplemental Information

Supplemental information for this article can be found online at http://dx.doi.org/10.7717/peerj.18306#supplemental-information.

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
