# Peer review of "Assessing acute effects of two motor-cognitive training modalities on cognitive functions, postural control, and gait stability in older adults: a randomized crossover study"

_PeerJ, doi:10.7717/peerj.18306_

## Round 0.1 · original submission · Minor Revisions

Firstly, I apologize for the time taken to select the reviewers.

I have completed my evaluation of your manuscript. As reviewers mentioned, I also consider this article interesting and sufficiently appropriate in providing additional and extended knowledge on the acute impact of different motor-cognitive training regimes on cognitive, function, postural, and walking ability in the elderly population. The reviewers recommend reconsideration of your manuscript following revision and modification.

Reviewer 1 ·

Basic reporting

no comment

Experimental design

no comment

Validity of the findings

no comment

Additional comments

The aim of the present study was to investigate the acute effects of two distinct motor-cognitive training modalities, treadmill dual-task training (TMDT) and interactive motor-cognition training on cognitive function, postural control, walking ability, and dual-task performance in the elderly population. The study is well-written, easy to read, and provides new insights into the potentiality of motor-cognitive training for physical and cognitive performance for counteracting the elderly individuals. Aging is often associated with a decline in cognitive function, postural control, and gait stability, which increases the risk of falls among older individuals. To address these challenges, the Authors propose a study on motor-cognitive training, a promising intervention to mitigate age-related declines. The findings suggest that interactive motor-cognitive training (performed with the use of Nintendo, in the form of game) induced greater immediate positive effect on cognitive function and gait stability among older adults compared to treadmill dual-task training, which seems to slightly impair postural control. This demonstrates the potential of interactive motor-cognitive training for mitigating fall risk and improving both cognitive and physical domains in the elderly population. The introduction offers a thorough examination of the research field related to physical and cognitive performance for ageing, with a specific emphasis on the practical applications for these types of training. Methods are complete and Results are well-presented.
Discussion is clear and provides and in-depth analysis of the findings in light of the literature. Overall, I would like to congratulate the Authors for their work that deserves publication.


I have only a couple of minor suggestions that I hope can contribute to complete the overall quality of the manuscript.
In the Introduction section, acute effects of exercise on cognitive functions are described. Emphasis is given to aerobic exercise. However, as other forms of exercises have been studied (this is in line with the combination of motor-cognitive training used in the present study), I would suggest to briefly introduce this concept suggesting the positive acute effects of balance and yoga exercises on cognitive performance. Here some suggested reference to refer:
Formenti, D., Cavaggioni, L., Duca, M., Trecroci, A., Rapelli, M., Alberti, G., Komar, J., Iodice, P., 2020. Acute Effect of Exercise on Cognitive Performance in Middle-Aged Adults: Aerobic Versus Balance. Journal of Physical Activity and Health 17, 773–780. https://doi.org/10.1123/jpah.2020-0005
Gothe, N., Pontifex, M.B., Hillman, C., McAuley, E., 2013. The Acute Effects of Yoga on Executive Function. Journal of Physical Activity and Health 10, 488–495. https://doi.org/10.1123/jpah.10.4.488

Moreover, I do believe that including possible practical applications derived from these findings might be helpful for the readers and also for practitioners working with older individuals (such as trainers, physical therapists, kinesiologists…). What are the practical implications and applications that can be derived from these findings?

Reviewer 2 ·

Basic reporting

I consider this article interesting and sufficiently appropriate in providing additional and extended knowledge on the acute impact of different motor-cognitive training regimes on cognitive, function, postural, and walking ability in the elderly population. However, the authors are invited to add more specific information over the methodological process in order to better reinforce data interpretation and support a potential study replication.

Here below additional specific comments
Intro
Line 58: replace “Consequently” to avoid redundant term within the surrounding text.
Line 76: instead of reporting an example over the Tai Chi, I would suggest exemplifying over the same subject (i.e., riding a bicycle)
Lines 123-124: IMO anticipating the results, it would not be the best strategy. Perhaps, at this stage, formulating a hypothesis would be appropriate.

Discussion
Lines 403-406: move this part later. Firstly, the main findings should be stated.
The discussion follows a logical structure and supports the results.
Please add the study limitations.

Experimental design

Lines 126-143
What about sample size calculation? Any a priori power analysis was conducted? Please clarify
Line 173: I believe that a figure representing the entire study design would favor the understating of the M&M section
Line 179: “RPE” instead of its extended form
Line 221: please clearly specify whether the familiarization was administered on the same day of the experimental session.
I believe reporting each testing reliability would be relevant? Did the authors check it?
Lines 320-323: add the effect size used for the mixed-anova (i.e., partial eta squared)
Line 323: “Should….” Please rephrase the sentence for clarity

Validity of the findings

no comment

·

Basic reporting

This manuscript seems to offer a comprehensive information on how social interaction and physical exercises affect the overall health and wellbeing of seniors. Some minor revisions are suggestable.

Authors should work closely with journal guidelines when submitting manuscripts. The whole manuscript for review should include the manuscript body document, figures, and tables in one single file. These files were disjointed for this work which made the review quite cumbersome. This should be corrected in case further revisions and resubmissions are requested.

Experimental design

Authors need to provide scientific bases and justifications for selections made at the design stages of the experimentations. For example, line 186-187 states "We implemented the Nintendo Ring Fit Adventure exergame as an IMCT intervention for subjects in our study." The question is why? Why was the Nintendo ring used? why not another? Any scientific references for this?

Revisions as this should be considered in the entire manuscript.

Validity of the findings

One striking question is this: Is/are there any association(s) between exercise and interaction when it comes to health of seniors? Authors seemed to have examined these factors independently for comparison on which is more effective.

---

## Round 0.2 · Minor Revisions

The authors should correct the use of ‘we’ and ‘our’ and Figure 1 in accordance with the comments from Reviewer 3.

Reviewer 1 ·

Basic reporting

The Authors have extensively addressed all my comments.

Experimental design

no comment

Validity of the findings

no comment

Additional comments

no comment

Reviewer 2 ·

Basic reporting

no comment

Experimental design

no comment

Validity of the findings

no comment

·

Basic reporting

The reporting seems to have been improved. One suggestable minor revision is the use of possessive pronouns such as "we" and "our" in the text body which is not too ethical for a technically constructed manuscript. This should be revised.

Experimental design

Figure 1 seems to offer a descriptive summary of the study which is commendable.

Validity of the findings

Findings seem to be improved.

---

## Round 0.3 · accepted · Accept

I confirmed that the authors have addressed all of the reviewer's comments.